# Yolk-sac-derived macrophages progressively expand in the mouse kidney with age

Shintaro Ide[1,2†], Yasuhito Yahara[2,3,4†], Yoshihiko Kobayashi[5], Sarah A Strausser[1], Kana Ide[1], Anisha Watwe[1], Shengjie Xu-Vanpala[6], Jamie R Privratsky[1], Steven D Crowley[1], Mari L Shinohara[6,7], Benjamin A Alman[2,3], Tomokazu Souma[1,2]*

[1]Division of Nephrology, Department of Medicine, Duke University School of Medicine, Durham, United States; [2]Regeneration Next, Duke University, Durham, United States; [3]Department of Orthopedic Surgery, Duke University School of Medicine, Durham, United States; [4]Department of Orthopedic Surgery, Faculty of Medicine, University of Toyama, Toyama, Japan; [5]Department of Cell Biology, Duke University School of Medicine, Durham, United States; [6]Department of Immunology, Duke University School of Medicine, Durham, United States; [7]Department of Molecular Genetics and Microbiology, Duke University School of Medicine, Durham, United States

*For correspondence:
tomokazu.souma@duke.edu

†These authors contributed equally to this work

Competing interests: The authors declare that no competing interests exist.

**Abstract** Renal macrophages represent a highly heterogeneous and specialized population of myeloid cells with mixed developmental origins from the yolk-sac and hematopoietic stem cells (HSC). They promote both injury and repair by regulating inflammation, angiogenesis, and tissue remodeling. Recent reports highlight differential roles for ontogenically distinct renal macrophage populations in disease. However, little is known about how these populations change over time in normal, uninjured kidneys. Prior reports demonstrated a high proportion of HSC-derived macrophages in the young adult kidney. Unexpectedly, using genetic fate-mapping and parabiosis studies, we found that yolk-sac-derived macrophages progressively expand in number with age and become a major contributor to the renal macrophage population in older mice. This chronological shift in macrophage composition involves local cellular proliferation and recruitment from circulating progenitors and may contribute to the distinct immune responses, limited reparative capacity, and increased disease susceptibility of kidneys in the elderly population.

## Introduction

Tissue-resident macrophages constitute a highly heterogeneous and specialized population of myeloid cells, reflecting the diversity of their developmental origins and tissue microenvironments (*Mass, 2018*; *Ginhoux et al., 2010*; *Schulz et al., 2012*; *Yona et al., 2013*; *Gomez Perdiguero et al., 2015*; *Mass et al., 2016*; *Hashimoto et al., 2013*; *Epelman et al., 2014*; *Stremmel et al., 2018*; *Hoeffel et al., 2015*). In addition to their critical roles in host defense against pathogens, macrophages are central to sterile inflammation, angiogenesis, and tissue remodeling, making them an attractive target for therapeutic intervention. Tissue-resident macrophages originate from at least three distinct progenitors: (*i*) macrophage colony-stimulating factor one receptor (CSF1R)-positive yolk-sac macrophages; (*ii*) CX3C chemokine receptor 1 (CX3CR1)-positive yolk-sac macrophages, also known as pre-macrophages; and (*iii*) embryonic and neonatal hematopoietic stem cells (HSC) (*Mass, 2018*; *Ginhoux et al., 2010*; *Schulz et al., 2012*; *Yona et al., 2013*; *Gomez Perdiguero et al., 2015*; *Mass et al., 2016*; *Hashimoto et al., 2013*; *Epelman et al., 2014*; *Stremmel et al.,*

**eLife digest** Older people are more likely to develop kidney disease, which increases their risk of having other conditions such as a heart attack or stroke and, in some cases, can lead to their death. Older kidneys are less able to repair themselves after an injury, which may help explain why aging contributes to kidney disease. Another possibility is that older kidneys are more susceptible to excessive inflammation. Learning more about the processes that lead to kidney inflammation in older people might lead to better ways to prevent or treat their kidney disease.

Immune cells called macrophages help protect the body from injury and disease. They do this by triggering inflammation, which aides healing. Too much inflammation can be harmful though, making macrophages a prime suspect in age-related kidney harm. Studying these immune cells in the kidney and how they change over the lifespan could help scientists to better understand age-related kidney disease.

Now, Ide, Yahara et al. show that one type of macrophage is better at multiplying in older kidneys. In the experiments, mice were genetically engineered to make a fluorescent red protein in one kind of macrophage. This allowed Ide, Yahara et al. to track these immune cells as the mice aged. The experiments showed that this subgroup of cells is first produced when the mice are embryos. They stay in the mouse kidneys into adulthood, and are so prolific that, over time, they eventually become the most common macrophage in older kidneys.

The fact that one type of embryonically derived macrophage takes over with age may explain the increased inflammation and reduced repair capacity seen in aging kidneys. More studies will help scientists to understand how these particular cells contribute to age-related changes in susceptibility to kidney disease.

---

*2018*). These populations are maintained in situ by self-renewal, largely independent of adult hematopoiesis (*Mass, 2018*; *Ginhoux et al., 2010*; *Schulz et al., 2012*; *Yona et al., 2013*; *Gomez Perdiguero et al., 2015*; *Mass et al., 2016*; *Hashimoto et al., 2013*; *Epelman et al., 2014*; *Hoeffel et al., 2015*; *Sawai et al., 2016*; *Soucie et al., 2016*).

Most tissues have mixed populations of different ontogenically-derived macrophages, and their relative contributions and temporal kinetics are tissue-specific. For example, microglia, Kupffer cells, and Langerhans cells originate from the yolk-sac, with minimal contribution from HSCs (*Ginhoux et al., 2010*; *Schulz et al., 2012*; *Yona et al., 2013*; *Gomez Perdiguero et al., 2015*; *Mass et al., 2016*; *Sawai et al., 2016*). The macrophage composition in the intestinal wall is highly dynamic. Yolk-sac-derived intestinal macrophages are rapidly replaced by HSC-derived macrophages after birth, but a subpopulation of yolk-sac-derived cells persist and self-renew in the specialized intestinal niches in adults (*Bain et al., 2014*; *De Schepper et al., 2018*). Importantly, investigators now recognize that macrophage ontogeny contributes to their roles in disease processes such as cancer progression; in pancreatic cancer, for example, yolk-sac-derived macrophages are fibrogenic, while HSC-derived macrophages are immunogenic (*Zhu et al., 2017*). This raises the possibility that developmental programs influence how macrophages differentially respond to disease insults.

Renal macrophages are found in an intricate network surrounding the renal tubular epithelium (*Stamatiades et al., 2016*; *Berry et al., 2017*; *Viehmann et al., 2018*) and have mixed origins from both yolk-sac and HSC (*Schulz et al., 2012*; *Mass et al., 2016*; *Epelman et al., 2014*; *Munro et al., 2019*). They exert unique functions depending on their anatomical locations; monitoring and clearing immune complexes is a function of cortical macrophages while bacterial immunity is the responsibility of medullary macrophages (*Stamatiades et al., 2016*; *Berry et al., 2017*). Renal macrophages critically control renal inflammation and tissue remodeling after injury with robust phenotypic reprogramming (*Lever et al., 2019*). Despite the importance of these roles, little is known about how the proportion and distribution of macrophages of different ontogeny change over time in the normal kidney and whether this influences the increased susceptibility and poorer outcomes of older patients to acute and chronic kidney diseases (*Chen et al., 2019*; *Chawla et al., 2014*; *Sato and Yanagita, 2019*). While most preclinical models of kidney diseases have used young animals, recent papers highlight distinct immune responses in aged mouse kidneys. Aged kidneys

exhibit more severe inflammation than young kidneys in response to ischemic and toxic insults, leading to maladaptive repair and organ dysfunction (*Sato and Yanagita, 2019*; *Sato et al., 2016*). Furthermore, there is a growing interest in aging as a fundamental determinant of macrophage heterogeneity, such as in the heart and serous cavities (*Molawi et al., 2014*; *Bain et al., 2016*; *Dick et al., 2019*). However, data on the effects of aging on the renal macrophage populations are lacking. Here, using complementary in vivo genetic fate-mapping and parabiotic approaches, we identify a previously unappreciated increase in the proportion of yolk-sac-derived macrophages in the mouse kidney with age.

## Results and discussion

Two complementary strategies were used to fate-map CSF1R$^+$, and CX3CR1$^+$ yolk-sac-derived macrophages (*Figure 1* and *Figure 1—figure supplement 1*). Erythromyeloid progenitors (EMP) give rise to these populations, and the yolk-sac macrophages appear in the yolk-sac around E8.5. Subsequently, from E9.0 until E14.5, they proliferate, migrate, and colonize the embryo through the vascular system (*Stremmel et al., 2018*; *Munro et al., 2019*). To lineage-label these cells, we exposed *Csf1r-CreERt*; *Rosa26$^{tdTomato}$* and *Cx3cr1$^{CreERt}$*; *Rosa26$^{tdTomato}$* embryos to 4-hydroxytamoxifen (4-OHT) at E8.5 and E9.5, respectively (*Figure 1A* and *Figure 1—figure supplement 1A*), (*Mass, 2018*). This efficiently and irreversibly labels yolk-sac-derived macrophages with the tdTomato reporter. Importantly, the approach does not label fetal monocytes or HSCs (*Gomez Perdiguero et al., 2015*; *Yahara et al., 2020*).

At postnatal day 0 (P0), we detected a small number of tdTomato$^+$ cells in kidneys from both lines (*Figure 1* and *Figure 1—figure supplement 1*). A previous fate-mapping strategy that labels all HSC-derived cells indicated that 40% to 50% of tissue-resident macrophages in the young adult kidney originate from HSC; the remainder was inferred to derive from yolk-sac hematopoiesis (*Schulz et al., 2012*). Consistent with this inference, we found CX3CR1-lineage labeled cells in kidneys from birth, with numbers increasing progressively over time (2 weeks, 2 months and 6 months; *Figure 1*, B and C). Surprisingly, we observed an unexpected large increase in the proportion of tdTomato-positive cells relative to total F4/80-positive cells at 6 months, especially in the cortex and outer medulla, despite no significant change in the number of F4/80-positive cells per section. This increase was maintained up to 1 year (*Figure 1B*). The tdTomato-labeled cells were positive for mature macrophage markers, F4/80 and CD64 (*Figure 1D*), (*Viehmann et al., 2018*; *Brähler et al., 2018*). By contrast, we observed only a few F4/80-positive CSF1R-lineage cells inside the kidneys (*Figure 1—figure supplement 1, B and C*), as reported previously (*Schulz et al., 2012*). As CSF1R$^+$ and CX3CR1$^+$ yolk-sac macrophages represent a developmental sequence of tissue-resident macrophages derived from EMP, the observed low labeling of CSF1R-lineage cells might be attributable to differences in labeling efficacies and migration kinetics of progenitors.

To further delineate the chronological shift of renal macrophages, we examined the HSC contribution to renal macrophages using the *Flt3-Cre; Rosa26$^{tdTomato}$* mouse line (*Mass, 2018*; *Yahara et al., 2020*; *Benz et al., 2008*). This mouse line irreversibly labels fetal and adult HSC-derived multipotent hematopoietic progenitors and their progeny with tdTomato expression. Consistent with the increase of yolk-sac-derived renal macrophages with age, we observed a decreased number of F4/80$^+$ tdTomato$^+$ cells in the kidneys from 6-month-old mice compared to those from 2-month-old mice (*Figure 2*, A and B). These data demonstrate that EMP-derived CX3CR1$^+$ yolk-sac macrophages and their descendants are major contributors to the resident renal macrophage population in aged kidneys.

Our findings raise the question of the underlying mechanisms responsible for the chronological shift of macrophage composition. We first tested whether the proliferation of yolk-sac-derived renal macrophages can potentially contribute to their population dynamics. CX3CR1-lineage-labeled cells that express the proliferation marker, Ki67, are present in kidneys from birth to 6 months of age, indicating that yolk-sac-derived macrophages retain the potential to expand in numbers through proliferation (*Figure 3*, A and B). We further found that higher percentages of CX3CR1-lineage-labeled F4/80$^+$ cells express Ki67 in comparison to tdTomato-negative F4/80$^+$ cells at 2 weeks and 2 months of age, suggesting that CX3CR1-lineage cells have a higher proliferating capacity (*Figure 3C*).

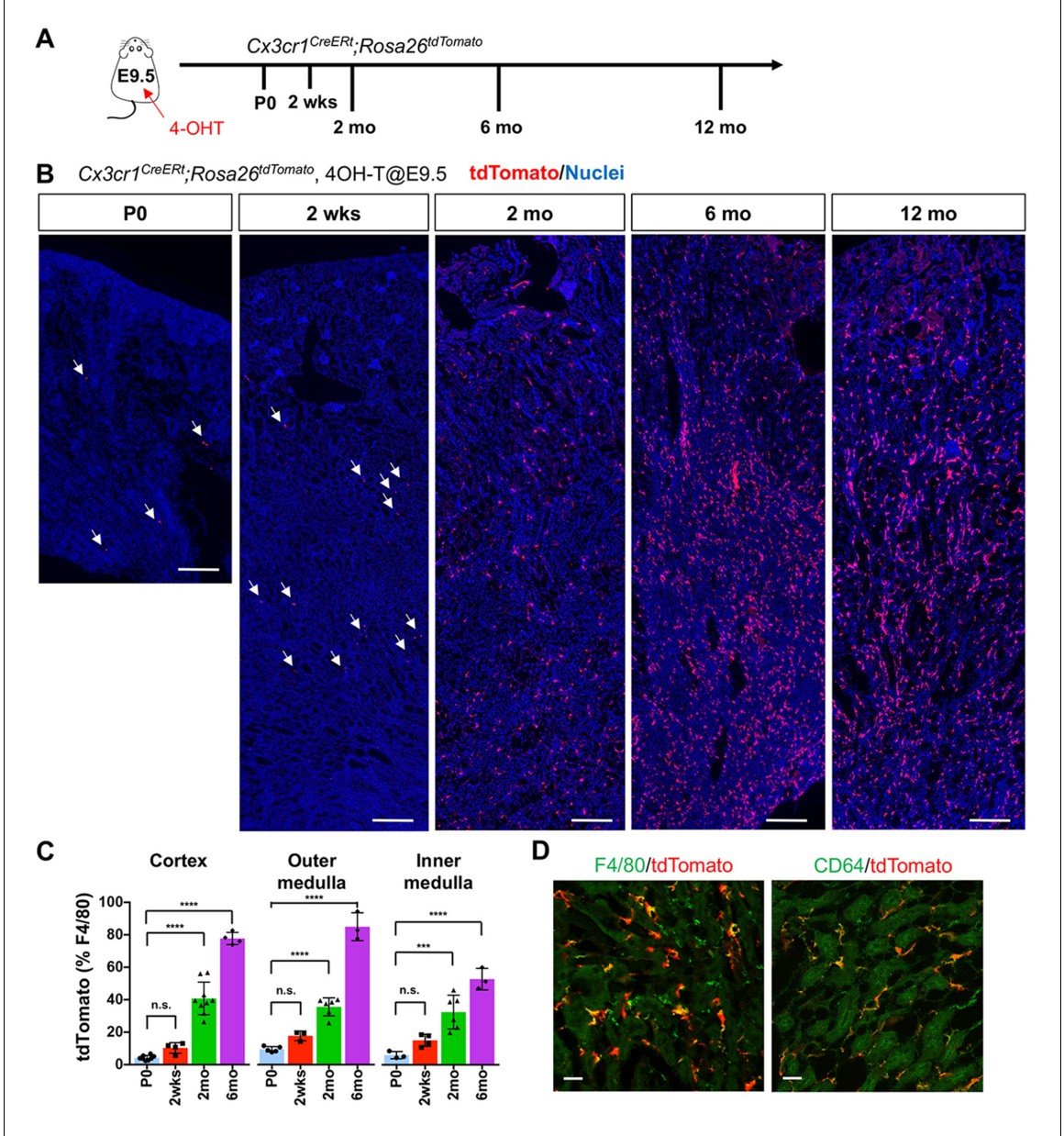

**Figure 1.** CX3CR1-positive yolk-sac macrophage descendants progressively expand in number in kidneys with age. (**A**) Fate-mapping strategies of CX3CR1$^+$ yolk-sac macrophages. 4-hydroxytamoxifen (4-OHT) was injected once into pregnant dams at 9.5 dpc and offspring analyzed at the indicated times (n = 4–6 for P0 to 6-month-old; n = 2 for 12-month-old). Yolk-sac macrophages and their progeny are irreversibly tagged with tdTomato. (**B**) Distribution of CX3CR1-lineage cells in postnatal kidneys. Arrows: CX3CR1-lineage cells. (**C**) Percentage of tdTomato$^+$ to F4/80$^+$ cells. Data are represented as means ± S.D. ***, p<0.001; ****, p<0.0001; n.s., not significant. (**D**) Confocal images of F4/80 and CD64 staining in aged kidneys (six mo) with CX3CR1-lineage tracing (n = 3). Scale bars: 200 µm in B; 20 µm in D.

The online version of this article includes the following source data and figure supplement(s) for figure 1:

**Source data 1.** Percentage of tdTomato+ to F4/80+ cells.
**Figure supplement 1.** CSF1R-positive yolk-sac macrophage descendants do not expand in number in kidneys.
**Figure supplement 2.** There is no basal Cre activity in kidneys without 4-hydroxytamoxifen (4-OHT) treatment.

Another possibility is recruitment of yolk-sac-derived macrophages from extra-renal reservoirs through the circulation. To test this hypothesis, we generated a parabiotic union between young *Cx3cr1*$^{GFP/+}$ and *Cx3cr1*$^{CreERt}$; *Rosa26*$^{tdTomato}$ mice that had been exposed to 4-OHT in utero at E9.5 (*Figure 4A*). Effective blood sharing between the pair was confirmed by detecting *Cx3cr1*-promoter-

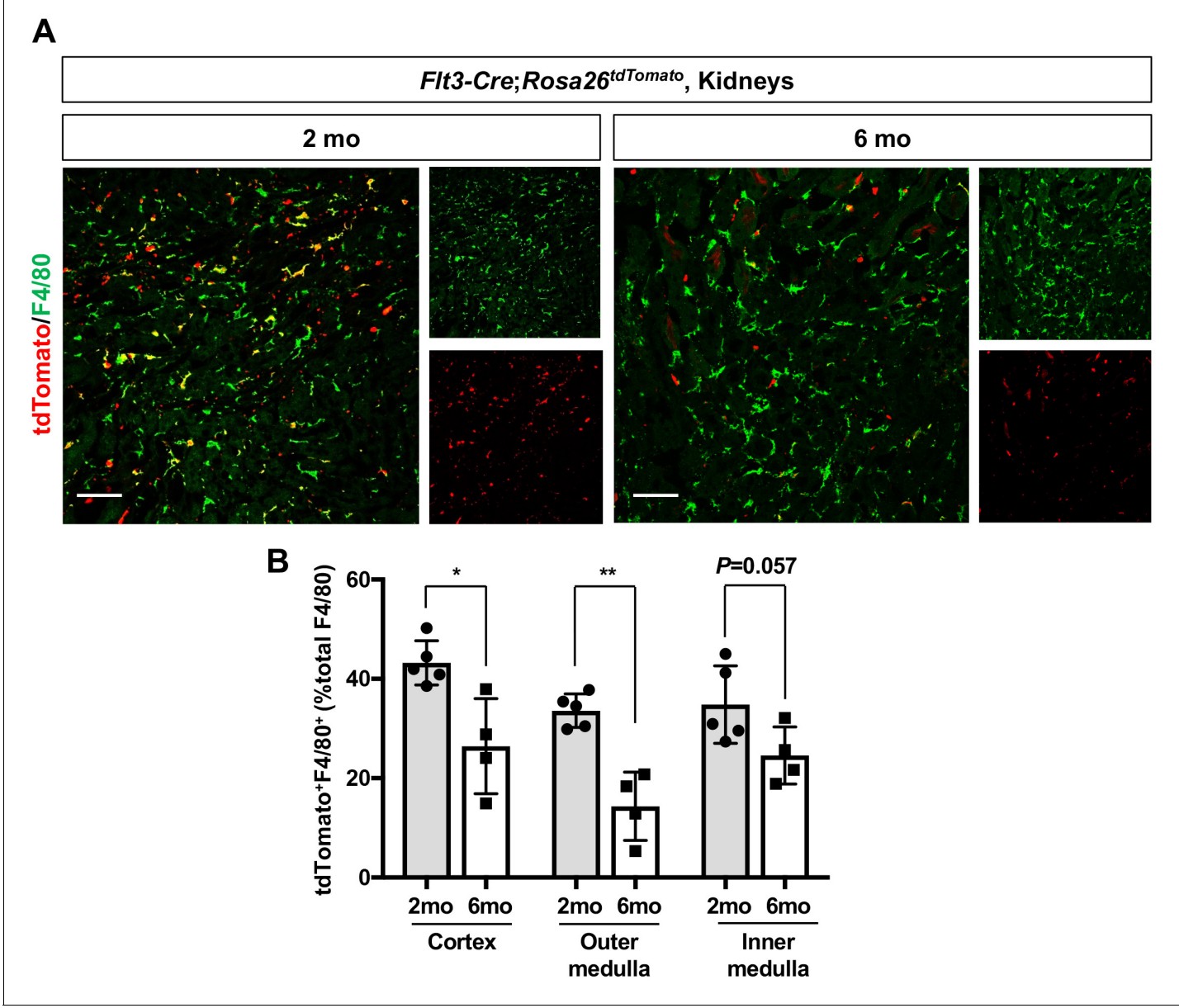

**Figure 2.** Age-dependent decrease of HSC-derived F4/80$^+$ cells in the kidneys. The *Flt3-Cre; Rosa26*$^{tdTomato}$ mouse line was used to examine the contribution of HSC-derived tissue-resident macrophages. (**A**) Distribution of F4/80$^+$ Flt3-lineage cells in postnatal kidneys. The kidneys were analyzed at the indicated times (n = 4–5). HSC-derived cells and their progeny are irreversibly tagged with tdTomato. (**B**) Percentage of tdTomato$^+$ F4/80$^+$ to total F4/80$^+$ cells. Note that the number of tdTomato$^+$ F4/80$^+$ cells decreases with age. Data are represented as means ± S.D. *, p<0.05; **, p<0.01. The online version of this article includes the following source data for figure 2:

**Source data 1.** Percentage of tdTomato+ F4/80+ cells to total F4/80+ cells.

driven GFP expression in bone marrow cells derived from both mice (data not shown). When analyzed at 5 weeks after parabiosis, we found a few tdTomato$^+$ cells in the extravascular, interstitial area of the cortex and medulla of the *Cx3cr1*$^{GFP}$ parabiont kidneys (0.105 ± 0.04% of total F4/80$^+$ cells; *Figure 4*, B and C). We also found that a significantly higher percentage of tdTomato-positive cells expresses Ki67 in the kidneys of *Cx3cr1*$^{GFP}$ mice compared to the tdTomato-positive cells in the kidneys of *Cx3cr1*$^{CreERt}$; *Rosa26*$^{tdTomato}$ mice (Ki67$^+$tdTomato$^+$ relative to tdTomato$^+$; 28.55 ±

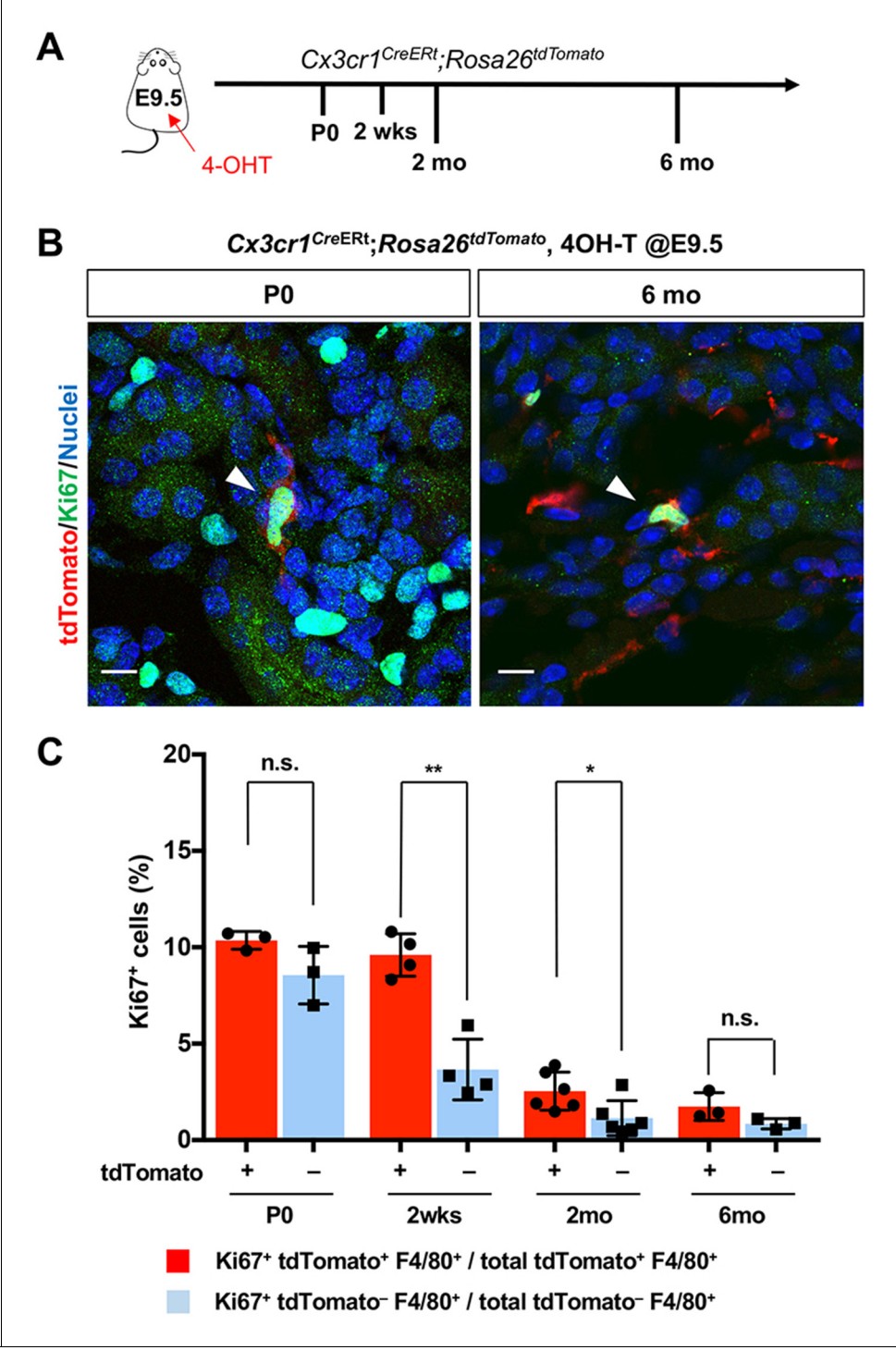

**Figure 3.** CX3CR1-positive yolk-sac macrophage descendants proliferate locally in the kidneys. (A) Schematic of fate-mapping strategy. (B and C) CX3CR1-lineage macrophages proliferate in neonatal and aged kidneys. $Cx3cr1^{CreERt}$; $Rosa26^{tdTomato}$ mice were treated with 4-hydroxytamoxifen (4-OHT) at E9.5 (n = 3–5). Arrowheads: Ki67$^+$ CX3CR1-lineage cells. Percentage of Ki67$^+$ proliferating cells are shown in C. Note that a higher percentage of CX3CR1-lineage F4/80$^+$ cells (tdTomato$^+$) are Ki67-positive compared to tdTomato$^-$ F4/80$^+$ cells. Data are represented as means ± S.D. *, p<0.05; **, p<0.01; n.s., not significant. Scale bars: 10 μm.

The online version of this article includes the following source data for figure 3:

**Source data 1.** Percentage of Ki67+ proliferating F4/80+ cells.

8.23% *vs.* 3.16 ± 0.74%) (*Figure 4*, D and E). While further investigation is required, these results suggest that the tdTomato-positive circulating progenitors may have a significant proliferative capacity and can slowly contribute to the adult renal macrophage pool. Currently, the origin of the circulating CX3CR1-lineage cells is not known but we speculate that one site is the spleen, which we recently identified as a reservoir of CX3CR1$^+$ yolk-sac macrophages (*Yahara et al., 2020*).

In conclusion, we have shown here that the proportion of yolk-sac-derived, CX3CR1-positive, macrophages increases significantly in the kidney with age, with recruitment from the circulation and proliferation being two possible mechanisms. Our findings provide a foundation for future studies to investigate the functional heterogeneity of ontogenically distinct renal macrophages in younger versus aged kidneys. These future studies may provide novel insight into age-related susceptibility of the kidney to acute and chronic diseases.

# Materials and methods

## Key resources table

| Reagent type (species) or resource | Designation | Source or reference | Identifiers | Additional information |
|---|---|---|---|---|
| Genetic Reagent (*M. musculus*) | Cx3cr1$^{CreERt}$ | The Jackson laboratory | RRID:IMSR_JAX:020940 | |
| Genetic Reagent (*M. musculus*) | Csf1r-CreERt (aka, Csf1r-Mer-iCre-Mer) | The Jackson laboratory | RRID:IMSR_JAX:019098 | |
| Genetic Reagent (*M. musculus*) | Flt3-Cre | | RRID:IMSR_EM:11790 | Flt3-Cre mice were bred into the C57BL6/J background for six generations by the Shinohara lab. |
| Genetic Reagent (*M. musculus*) | Rosa26$^{tdTomato}$ | The Jackson laboratory | RRID:IMSR_JAX:007914 | |
| Genetic Reagent (*M. musculus*) | Cx3cr1$^{GFP}$ | The Jackson laboratory | RRID:IMSR_JAX:005582 | |
| Antibody | Anti-F4/80 (Rat monoclonal) | Bio-Rad (MCA497) | RRID:AB_2098196 | Clone C1:A3-1 IF: 1:100 |
| Antibody | Anti-CD64 (Rat monoclonal) | Bio-Rad (MCA5997) | RRID:AB_2687456 | Clone AT152-9 IF: 1:200 |
| Antibody | Anti-Endomucin (Rat monoclonal) | Abcam (ab106100) | RRID:AB_10859306 | Clone V.7C7.1 IF: 1:100 |
| Antibody | Anti-Ki67 (Rat monoclonal) | eBioscience (14-5698-82) | RRID:AB_10854564 | Clone SolA15 IF: 1:200 |
| Antibody | Anti-Ki67 (Rabbit monoclonal) | Thermo (MA5-14520) | RRID:AB_10979488 | Clone SP6 IF: 1:200 |
| Antibody | Anti-dsRed (Rabbit polyclonal) | Rockland (600-401-379) | RRID:AB_2209751 | IF: 1:200 |
| Software, algorithm | ImageJ | NIH, Bethesda, MD (Version 1.52P) | RRID:SCR_003070 | https://imagej.nih.gov/ij/ |
| Software, algorithm | GraphPad Prism | | RRID:SCR_002798 | https://www.graphpad.com/scientific-software/prism/ |

## Study approval

All experiments were performed according to IACUC-approved protocols (A051-18-02 and A196-16-0).

## Animals

The mouse lines were from the Jackson Laboratory (Stock No: 019098; 020940; 007914; and 005582). The Flt3-Cre mouse line was kindly provided from Dr. K Lavine (Washington University, St. Louis, MO). 75 µg/g body weight of 4-hydroxytamoxifen (4-OHT; Sigma Aldrich, St. Louis, MO)

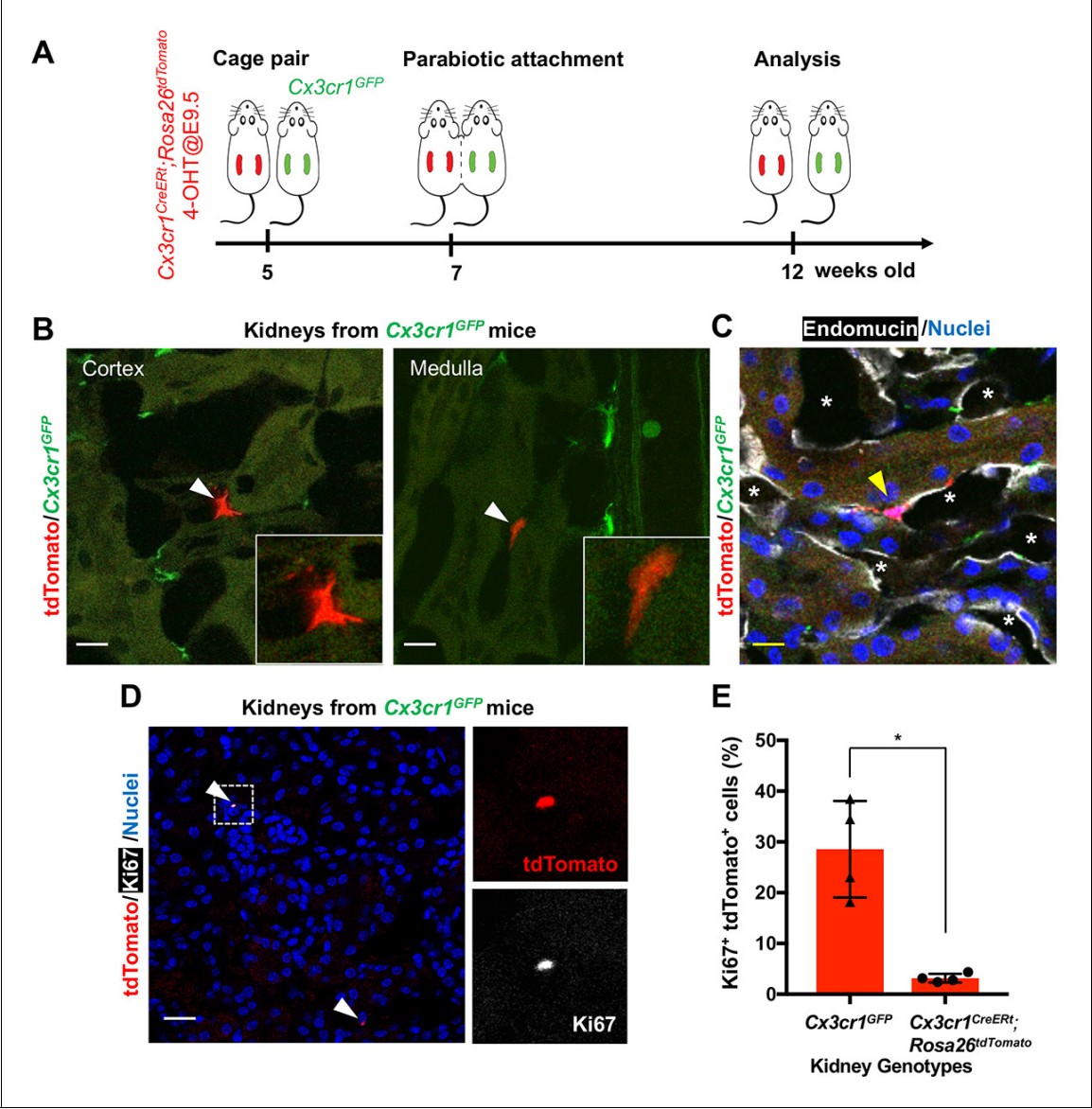

**Figure 4.** CX3CR1-positive yolk-sac macrophage descendants are recruited into adult kidneys from the circulation. (**A**) Schematic of parabiotic experiments. (**B and C**) Localization of tdTomato+ CX3CR1-lineage cells in *Cx3cr1*GFP kidneys. The tdTomato-positive cells were lineage-labeled in *Cx3cr1*CreERt; *Rosa26*tdTomato mice in utero at E9.5. They migrated into the parabiont Cx3cr1GFP kidneys from circulation. Note that tdTomato+ cells were detected in extravascular interstitium (n = 4 per group). Endomucin; an endothelial cell marker. *, lumen of capillaries. Arrowheads, CX3CR1-lineage cells from circulation. (**D and E**) Circulation-derived CX3CR1-lineage cells proliferate in adult kidneys. Arrowheads, CX3CR1-lineage cells from circulation. GFP fluorescence was lost during the antigen retrieval process to detect Ki67. Percentages of Ki67+tdTomato+ cells relative to tdTomato+ cells in the kidneys of indicated genotype are shown in E (n = 4 per group). Note that tdTomato+ cells in *Cx3cr1*GFP kidneys are derived from circulation. Data are represented as means ± S.D. *, p<0.05. Scale bars: 10 μm in B and C; and in 20 μm in D. Legends for the Supplementary Figures. The online version of this article includes the following source data and figure supplement(s) for figure 4:

**Source data 1.** Percentages of Ki67+tdTomato+ cells relative to tdTomato+ cells in the kidneys of indicated genotypes.

**Figure supplement 1.** There is no basal Cre activity in kidneys without 4-hydroxytamoxifen (4-OHT) treatment in the spleen of 2-month-old-mice.

dissolved in corn oil (Sigma Aldrich) was intraperitoneally administered into pregnant dams with 37.5 μg/g body weight progesterone (Sigma Aldrich) to avoid fetal abortions. Mice without 4-OHT treatment were used for the specificity of tdTomato signals (*Figure 1—figure supplement 2* and *Figure 4—figure supplement 1*). Animals were allocated randomly into experimental groups and

analyses. To determine experimental group sizes, data from our previous study were used to estimate the required numbers.

## Parabiosis surgery

Parabiosis surgery was performed as previously described (*Yahara et al., 2020*). Briefly, a longitudinal skin incision was performed from elbow to knee joint to each mouse. The two mice were connected by suture. Parabionts were separated 5 weeks after the surgery. The mice were euthanized, and kidneys and spleen were harvested for analyses. Sham surgery was performed in the same way except for joining two animals.

## Antibodies and sample processing

Primary antibodies: F4/80 (Bio-Rad, Hercules, CA; clone Cl:A3-1), CD64 (Bio-Rad; clone AT152-9), Endomucin (Abcam; Cambridge, UK; clone V.7C7.1), Ki67 (eBioscience, San Diego, CA; clone SolA15 and Thermo, Waltham, MA; clone SP6), and dsRed (Rockland, Limerick, PA; cat. #600-401-379). Fluorescent-labeled secondary antibodies were used appropriately. 7 μm cryosections were stained using standard protocols. Heat-induced antigen retrieval was performed using pH6.0 sodium citrate solution (eBioscience). Images were captured using Axio imager and 780 confocal microscopes (Zeiss, Oberkochen, Germany). More than three randomly selected areas from 3 to 5 kidneys were imaged and quantified using ImageJ.

## Statistics and reproducibility

Results are expressed as means ± SD. Unpaired t-test was used for comparing two groups. One-way ANOVA followed by Dunnett's correction was used for multiple group comparison. A P-value less than 0.05 was considered statistically significant.

## Acknowledgements

This study was supported by grants from the National Institute on Aging R01 AG049745 to BA, and NIH-AI088100 to MLS, and the American Society of Nephrology Career Developmental Grant and Duke Nephrology Start-up Fund to TS. SI and YY are supported in part by fellowship grants from the American Heart Association and the Kanzawa Medical Research Foundation, respectively. We thank Drs. Brigid Hogan, Benjamin Thomson, and Myles Wolf for comments and helpful suggestions on the manuscript. We also thank Mr. Puviindran Nadesan for his technical support. Imaging was performed at the Duke Light Microscopy Core Facility supported by the shared instrumentation grant (1S10RR027528-01).

## Additional information

### Funding

| Funder | Grant reference number | Author |
|---|---|---|
| American Society of Nephrology | Career Developmental Grant | Tomokazu Souma |
| Duke University School of Medicine | Start-up Fund | Tomokazu Souma |
| National Institute on Aging | R01 AG049745 | Benjamin A Alman |
| National Institute of Allergy and Infectious Diseases | AI088100 | Mari L Shinohara |
| American Heart Association | Postdoctoral fellowship | Shintaro Ide |
| Kanzawa Medical Research Foundation | Fellowship | Yasuhito Yahara |

The funders had no role in study design, data collection and interpretation, or the decision to submit the work for publication.

## Author contributions

Shintaro Ide, Data curation, Formal analysis, Investigation, Writing - review and editing; Yasuhito Yahara, Conceptualization, Resources, Investigation, Methodology, Writing - review and editing; Yoshihiko Kobayashi, Steven D Crowley, Resources, Methodology, Writing - review and editing; Sarah A Strausser, Investigation, Project administration; Kana Ide, Formal analysis, Investigation, Project administration; Anisha Watwe, Investigation; Shengjie Xu-Vanpala, Resources, Methodology; Jamie R Privratsky, Resources, Investigation, Methodology, Writing - review and editing; Mari L Shinohara, Resources, Funding acquisition, Methodology, Writing - review and editing; Benjamin A Alman, Resources, Supervision, Funding acquisition, Methodology, Writing - review and editing; Tomokazu Souma, Conceptualization, Supervision, Funding acquisition, Investigation, Writing - original draft, Project administration, Writing - review and editing

## Author ORCIDs

Shintaro Ide (iD) https://orcid.org/0000-0002-9301-211X
Yoshihiko Kobayashi (iD) https://orcid.org/0000-0001-7031-1478
Kana Ide (iD) https://orcid.org/0000-0002-2845-8481
Shengjie Xu-Vanpala (iD) https://orcid.org/0000-0003-2716-6230
Jamie R Privratsky (iD) https://orcid.org/0000-0003-3598-4911
Steven D Crowley (iD) https://orcid.org/0000-0002-1838-0561
Mari L Shinohara (iD) https://orcid.org/0000-0002-6808-9844
Tomokazu Souma (iD) https://orcid.org/0000-0002-3285-8613

## Ethics

Animal experimentation: This study was performed in strict accordance with the recommendations in the Guide for the Care and Use of Laboratory Animals of the National Institutes of Health. All animals were used according to the approved protocols (A051-18-02 and A196-16-0) by the Institutional Animal Care and Use Committee of Duke University.

## Decision letter and Author response

Decision letter https://doi.org/10.7554/eLife.51756.sa1
Author response https://doi.org/10.7554/eLife.51756.sa2

# Additional files

## Supplementary files

• Transparent reporting form

## Data availability

All data generated or analyzed during this study are included in the manuscript.

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
