## [Decision Letter]

**Acceptance summary:**

Your study provides compelling evidence of an age-dependent expansion of kidney macrophages, and the first evidence of a yolk-sac derived circulating macrophage progenitor. Your findings will be of high interest for the macrophage community.

**Decision letter after peer review:**

Thank you for submitting your article "Yolk-sac-derived macrophages progressively expand in the mouse kidney with age" for consideration by *eLife*. Your article has been reviewed by two peer reviewers, and the evaluation has been overseen by a Reviewing Editor and Satyajit Rath as the Senior Editor. The following individual involved in review of your submission has agreed to reveal their identity: Elvira Mass (Reviewer #1).

The reviewers have discussed the reviews with one another and the Reviewing Editor has drafted this decision and identified the major revisions needed to help you prepare a revised submission. The detailed comments of the reviewers are also provided for ease of reference.

Essential revisions:

Below are comments highlighting the important points to consider:

1) The control experiments requested by reviewer #1 should be carefully addressed.

2) Further evidence of local proliferation versus recruitment as requested by reviewer #2 should be provided to strengthen the manuscript.

3) Potential functional implication/s of YS-MF expansion in the kidney should be investigated to highlight the novelty of the current manuscript compared to the parallel submission suggesting recruitment from the spleen.

Reviewer #1:

Ide et al. characterize the origin of renal macrophages during aging and provide evidence of a yolk-sac derived macrophage precursor that persists in adult life. Using 2 different fate-mapping models they show that pre-macrophages (Cx3cr1+ at E9.5) give rise to a small proportion of resident macrophages in the kidney, which will expand throughout aging. The expansion can be explained by proliferation of yolk-sac derived cells and/or a circulating macrophage precursor originating in the spleen, which continuously contributes to the pool of macrophages as shown by parabiosis experiments.

The authors provide a compelling data set of an age-dependent expansion of kidney macrophages, and the first evidence of a yolk-sac derived circulating macrophage progenitor. Especially that latter is of high interest for the macrophage community. However, to ascertain both the expansion and the presence of a circulating progenitor in adult and aging mice, some experiments need to be performed/reanalyzed.

1) Control experiments with Cx3cr1^CreER^; Rosa26^tdTomato^ adult mice (not only P3) that have not been injected with tamoxifen should be performed (including parabiosis) to make sure that there is no leakiness of the reporter over time (Fonseca et al. 2017, J. Neuroinflammation). Particularly, the tdTomato driver is prone to leakiness, therefore another reporter could also be considered (e.g. dsRed or YFP).

2) Figure 3B: the authors claim that they see tdTomato signal in GFP+ macrophages of the Cx3cr1-GFP parabiont. It is however difficult to see an overlap of the signals. Representation of single colours would be helpful. Further, assessment of the% as done for Figure 1 would give the reader a better understanding of how often macrophage progenitors replace resident macrophages during adulthood.

Reviewer #2:

The current notion is that, with the notable exception of the CNS MF, HSC-derived cells seed peripheral organs with age and replace original YS-derived MF macrophages. The finding of Ide et al. that following this initial dilution of YS-MF there can be a resurgence of YS-derived kidney macrophages is provocative and definitively deserves attention. The source of these cells merits in depth study, be it that they arise from local proliferation, the recruitment from a splenic YS-MF depot, or even from EMP / YS-MF precursors persisting into adulthood that might so far have missed attention. As such this study is very interesting.

Specifically, a key observation and potentially conceptual advance of the present manuscript is that the late YS MF burst might derive from a splenic YS-MF infiltrate. This is supported by the parabiosis experiment, though experimentally not strictly dissected from local proliferation. If this aspect could be further carved out during a revision this would significantly strengthen the paper.

To support the stringency their experimentation but avoid duplication, the authors included a manuscript that is under revision elsewhere, and shows convincing proper controls.

However, this study by Yahara et al. also suggests the existence of splenic YS-macrophages that in this case are mobilized for osteoclast generation. This considerably compromises novelty of the study by Ide et al. To balance this shortcoming, and justify a re-publication of this finding in another organ, the authors would I believe have to include evidence for a specific functional implication of the YS-MF expansion for the kidney. As is the study feels more like an add-on of the Yahara et al. study that hardly stands on its own, not the least because it heavily relies on a yet unpublished paper for proper judgement.

[Editors' note: further revisions were suggested prior to acceptance, as described below.]

Thank you for resubmitting your work entitled "Yolk-sac-derived macrophages progressively expand in the mouse kidney with age" for further consideration by *eLife*. Your revised article has been evaluated by Satyajit Rath as the Senior Editor, a Reviewing Editor and two peer reviewers.

The manuscript has been improved but there are some remaining issues that need to be addressed before acceptance, as outlined below:

1) The new Figure 4D, E: this staining seems to be unspecific. The authors have to include a macrophage marker such as Iba1 or F4/80 to make sure that the Ki67/tdTomato signal stems from a macrophage.

Add the word 'local' to the last sentence of the Abstract "This chronological shift in macrophage composition involves local cellular proliferation and recruitment from circulating progenitors and may contribute to the distinct immune responses, limited reparative capacity, and increased disease susceptibility of kidneys in the elderly population."

---

## [Author Response]

Reviewer #1:[…] The authors provide a compelling data set of an age-dependent expansion of kidney macrophages, and the first evidence of a yolk-sac derived circulating macrophage progenitor. Especially that latter is of high interest for the macrophage community. However, to ascertain both the expansion and the presence of a circulating progenitor in adult and aging mice, some experiments need to be performed/reanalyzed.1) Control experiments with Cx3cr1^CreER^; Rosa26^tdTomato^ adult mice (not only P3) that have not been injected with tamoxifen should be performed (including parabiosis) to make sure that there is no leakiness of the reporter over time (Fonseca et al. 2017, J. Neuroinflammation). Particularly, the tdTomato driver is prone to leakiness, therefore another reporter could also be considered (e.g. dsRed or YFP).

We thank reviewer 1 for raising this important point. We have performed control experiments, as suggested. We also performed multiple experiments, which include two additional protocols to fate map the first wave of pre-macrophages (4-OH tamoxifen injection at E8.5) and hematopoietic stem cell-derived population (Flt3-Cre; Rosa26^tdTomato^). The new data summarized below strongly support our original notion that yolk-sac derived macrophages expand with age in kidneys.

1) No tamoxifen control. In our recent cohort, we did not observe any basal Cre activity without tamoxifen injection in the spleen, liver, and kidneys of 2-month-old Cx3cr1^CreERt^; Rosa26^tdTomato^ mice (0 out of 3 biological replicates: See Figure 1—figure supplement 1C). Our results are consistent with the recent report that showed our Cx3cr1^CreERt^(Jung); Rosa26^tdTomato^ mouse line, which was established by Dr. Steffen Jung, is significantly less leaky compared to another commonly used mouse line Cx3cr1^CreERT2^ (WganJ); Rosa26^tdTomato^ (Dick SA et al., 2019).

2) No tamoxifen control (Sham surgery group). We further tested whether surgical procedures activate the Cre activity without tamoxifen injection. While we did not observe any Cre basal activity in 2 out of 3 mice tested, we observed a moderate Cre activity in one of the animals (Figure 4—figure supplement 1). In contrast, we observed robust labeling of CX3CR1-lineage cells in all the spleens from 4-OH tamoxifen-treated animals (7 out of 7 spleens, 4-OH tamoxifen was given at E9.5). As shown in Figure 4, we also observed consistent and reproducible results when we analyzed the kidneys of parabiotic animals. Importantly, we observed that the circulation-derived CX3CR1-lineage cells possess a profound proliferative capacity (Figure 4E). Collectively, our data demonstrate that our lineage labeling strategy can effectively identify a yolk-sac derived circulating macrophage progenitor population.

3) Fate mapping of the first wave of pre-macrophages (CX3CR1^+^ yolk-sac macrophages). We lineage labeled the first wave of pre-macrophages using Cx3cr1^CreERt^; Rosa26^tdTomato^ mouse line. We injected 4-OH tamoxifen at 8.5 dpc into the pregnant dam and harvested the kidneys at 2 months after birth. We observed significantly less tdTomato-labeled cells in the kidneys of 2-month-old animals compared to those with E9.5 labeling (Author response image 1). These data demonstrate that our lineage labeling strategies effectively fate map different stages of premacrophages.

**Author response image 1. respfig1:** Comparison of fate-mapping strategies using different 4-OH tamoxifen treatment protocols. 4-hydroxytamoxifen (4-OHT) was injected once into pregnant dams at 8.5 dpc or 9.5 dpc, and the offspring were analyzed at 2 months of age. The first wave of pre-macrophages can be lineage traced with 4-OH tamoxifen injection at 8.5 dpc.

Note that E9.5 4-OH tamoxifen treatment more efficiently fatemaps the pre-macrophage-derived cells in the 2-month-old kidneys (5-fold increase of tdTomato-positive cells). Image-J was used for quantification. DAPI was used for nuclear staining.

4) Hematopoietic stem cell (HSC)-derived macrophages. We further tested whether the renal macrophages of HSC origin decrease with age using the Flt3-Cre; Rosa26^tdTomato^ mouse line (Mass, 2018; Benz et al., 2008). We observed a decreased number of F4/80^+^ tdTomato^+^ cells in the kidneys from 6-month-old animals compared to those from 2-month-old animals (see Author response image 2). This data mirrors our data of YS-derived macrophages (see Figure 1) and supports our original finding of the dynamic temporal changes of renal macrophage composition.

**Author response image 2. respfig2:** Age-dependent decline of HSC-derived F4/80-positive cells in kidneys. The Flt3-Cre; Rosa26^tdTomato^ mouse line was used to delineate the contribution of HSC-derived tissue resident macrophages. Please note that the number of tdTomato^+^ F4/80^+^ cells (yellow), which indicates HSC origin, decreases with age. *P<0.05, **P<0.01. Image-J was used for quantification. Flt3-Cre mice were kindly provided by Dr. K. Lavine (Washington University, St. Louis MO). Flt3-Cre mice were bred into the C57BL6/J background for 6 generations by the Shinohala lab.

5) No tdTomato^+^ cells in the bone marrow CD45^+^ cells. We have recently published that our lineage labeling strategy did not label CD45^+^ cells in the bone marrow of adult animals at 2 months old (Yahara et al., 2020), while there are many CX3CR1-positive cells in this population. We extended these analyses to 6 month-old animals, which were exposed to 4-OH tamoxifen in utero at E9.5. We found that there was “no increase” of tdTomato labeled cells in the CD45^+^ cells over time (biological replicate n=4, Author response image 3).

We obtained kidney tissues from this cohort of animals for Figure 1. These data further support our original contention that our fate-mapping strategy is selective and efficient.

**Author response image 3. respfig3:** No increase of CD45^+^ tdTomato^+^ cells in bone marrow of 6 month-old Cx3cr1^CreERt^; Rosa26^tdTomato^ mice. We analyzed thenumber of tdTomato^+^ CD45^+^ cells in bone marrow (N=4). Compared to 2-month-old bone marrow (0.02% of CD45^+^ cells, Yahara et al., 2020), the number of tdTomato^+^CD45^+^ cells did not increase with age (0.025%).

2) Figure 3B: the authors claim that they see tdTomato signal in GFP+ macrophages of the Cx3cr1-GFP parabiont. It is however difficult to see an overlap of the signals. Representation of single colours would be helpful. Further, assessment of the% as done for Figure 1 would give the reader a better understanding of how often macrophage progenitors replace resident macrophages during adulthood.

We agree with the reviewer that the images and descriptions were unclear. In our parabiosis experiments, we created parabiotic unions for 5 weeks between Cx3cr1^CreERt^; Rosa26^tdTomato^ mice (exposed to 4-OH tamoxifen at E9.5) and Cx3cr1^GFP^ mice. To identify the contribution of circulation derived macrophage progenitors of YS origin in renal macrophage homeostasis, we analyzed the kidneys from Cx3cr1^GFP^ mice. We considered tdTomato-lineage-labeled cells in the kidneys of Cx3cr1^GFP^ mice as circulation-derived cells. As suggested by this reviewer, we provide single-color images of Figure 4B (see Author response image 4).

We also quantified the tdTomato-labeled cells (0.105 ± 0.04% of total F4/80-positive cells). While the number of circulation-derived tdTomato^+^ cells are low, they showed a significantly high proliferation rate (see Figure 4E).

**Author response image 4. respfig4:** Merged and single color panels of Figure 4B. The tdTomato-positive cells in the Cx3cr1^GFP^ kidneys are derived from the parabiont animals (Cx3cr1^CreERt^; Rosa26^tdTomato^) through circulation. GFP-positive cells are resident Cx3cr1-expressing cells.

Reviewer #2:[…] A key observation and potentially conceptual advance of the present manuscript is that the late YS MF burst might derive from a splenic YS-MF infiltrate. This is supported by the parabiosis experiment, though experimentally not strictly dissected from local proliferation. If this aspect could be further carved out during a revision this would significantly strengthen the paper.

We thank this reviewer for positive and constructive comments to improve our manuscript in clarity and novelty. As suggested by this reviewer, we investigated the proliferative capacity of yolk-sac-derived macrophages. We observed the highest proliferative capacity of circulating progenitor-derived macrophages (28.55 ± 8.23%) compared to total CX3CR1-lineage cells (3.16 ± 0.74%) and other lineage macrophages (1.13 ± 0.85%) in the adult kidneys (see Author response image 5, and Figure 3 and 4).

**Author response image 5. respfig5:** Circulation-derived CX3CR1-lineage cells have a profound proliferative capacity. The percentage of Ki67^+^ cells is shown. Data are represented as means ± S.D. The mice were exposed to 4-OH tamoxifen in utero at E9.5. The tdTomato^+^ CX3CR1-lineage cells are descendants of yolk-sac macrophages. Data are derived and summarized from Figures 3 and 4 (see representative images in these figures). Progenitor, circulation-derived CX3CR1-lineage cells identified in Cx3cr1^GFP^ kidneys; Total CX3CR1-lineage cells, macrophages derived from CX3CR1^+^ yolk-sac macrophages in the kidneys of Cx3cr1^CreERt^; Rosa26^tdTomato^ mice; Other lineages, F4/80^+^tdTomato^–^ macrophages in the kidneys of Cx3cr1^CreERt^; Rosa26^tdTomato^ mice. Image-J was used for quantification.

To support the stringency their experimentation but avoid duplication, the authors included a manuscript that is under revision elsewhere, and shows convincing proper controls.However, this study by Yahara et al. also suggests the existence of splenic YS-macrophages that in this case are mobilized for osteoclast generation. This considerably compromises novelty of the study by Ide et al. To balance this shortcoming, and justify a re-publication of this finding in another organ, the authors would I believe have to include evidence for a specific functional implication of the YS-MF expansion for the kidney. As is the study feels more like an add-on of the Yahara et al. study that hardly stands on its own, not the least because it heavily relies on a yet unpublished paper for proper judgement.

We thank this reviewer for carefully reviewing our manuscripts. The related manuscript by Yahara et al. is now published in Nature Cell Biology (Yahara et al., 2020). As this reviewer pointed out, we reported the potential contribution of YS-derived macrophage progenitors in “bone” homeostasis and repair. However, we did not know whether the identified YS-derived macrophage progenitors contribute to homeostasis of tissue-resident macrophages in visceral organs as the bone is a highly specialized unique organ, which is mineralized.

In the current manuscript, we could identify the progenitor-derived macrophages of circulation origin in adult kidneys. We also found that these progenitors possess impressive proliferative capacity (Figure 4D and E; Author response image 5). Our result supports the novel concept that “yolk-sac-derived macrophage progenitors circulate and contribute to tissue-resident macrophage pool in diverse niches in adults”. Our data may be widely generalized to tissue macrophage homeostasis in other visceral organs.

We also believe our report has significant biological and clinical implications. Clinically, aging is one of the major risk factors of kidney diseases, which afflict around 10% of the population worldwide. A recent elegant report highlighted that renal inflammatory responses after ischemia-reperfusion injury are primarily dependent on the age of mice (Sato et al., 2016). They showed that aged kidneys exhibit significantly more severe inflammation and limited renal repair compared to the young. The aged kidneys produced inflammatory cytokines (ex. osteopontin and Tnf-α), which are known to be produced by macrophages, at higher levels compared to those of young kidneys (Sato et al., 2016).

Our observed age-dependent chronological shift in macrophage composition may explain these distinct immune responses, limited organ reparative ability, and poor outcomes of aged kidneys. We would like to address these exciting questions as future studies. We are aware of the *eLife*’s excellent mechanism, “Research Advances”, which is intended to publish following-up findings after a publication in *eLife*. We hope to publish future findings as a Research Advances in *eLife*.

[Editors' note: further revisions were suggested prior to acceptance, as described below.]

The manuscript has been improved but there are some remaining issues that need to be addressed before acceptance, as outlined below:1) The new Figure 4D, E: this staining seems to be unspecific. The authors have to include a macrophage marker such as Iba1 or F4/80 to make sure that the Ki67/tdTomato signal stems from a macrophage.

We thank the editors for raising this important point. As mentioned above, we rephrased the high potential of circulating CX3CR1-lineage progenitors less conclusive.

Regarding the septicity of the signal, we have the following supporting evidence that shows the tdTomato signal in Figure 4D and E is specific to “CX3CR1-lineage cells”, as we communicated in the previous letter.

– We observed a morphological variation of CX3CR1-lineage cells in the parabiont Cx3cr1^GFP^ kidneys from oval to the elongated shape (see Author response image 6). This data is consistent with our lineage-tracing analyses in Figures 1 and 3, as we also observed the similar morphological variations in these kidneys (see Figure 3B for high magnification).

– Many of Ki67-positive CX3CR1-lineage cells in parabiont Cx3cr1^GFP^ kidneys are oval-shaped, but Ki67-negative ones have elongated shapes. We speculate that the circulation-derived CX3CR1-lineage cells spread their processes to survey the renal environment after proliferation in situ.

– The oval CX3CR1-lineage cells are also stained with DAPI, excluding a possibility that the signal is an artifact (see Author response image 6).

– We did not observe any tdTomato-positive cells in negative control kidneys without tamoxifen injection.

**Author response image 6. respfig6:** ****Morphological variation of CX3CR1-lineage cells. The CX3CR1-lineage tdTomato-positive cells in the Cx3cr1^GFP^ kidneys are derived from the parabiont animals (Cx3cr1^CreERt^; Rosa26^tdTomato^) through circulation. The Cx3cr1^CreERt^; Rosa26^tdTomato^ mice were exposed to 4-OH tamoxifen in utero at E9.5. (**A**) Ki67^+^ CX3CR1-lineage cells are positive for DAPI (nuclear staining). (**B**) The CX3CR1-lineage tdTomato-positive cells exhibit oval and elongated shapes. Scale bars: 2 µm in A and 5 µm in B.

Add the word 'local' to the last sentence of the Abstract "This chronological shift in macrophage composition involves local cellular proliferation and recruitment from circulating progenitors and may contribute to the distinct immune responses, limited reparative capacity, and increased disease susceptibility of kidneys in the elderly population."

We have added the word “local” accordingly.